# Filtration of Nutritional Fluids in the German Wasp *Vespula germanica* (Vespidae, Hymenoptera)

**DOI:** 10.3390/insects13020185

**Published:** 2022-02-10

**Authors:** Kenneth Kuba, Harald W. Krenn

**Affiliations:** 1Department of Evolutionary Biology, Integrative Zoology, University of Vienna, 1030 Vienna, Austria; harald.krenn@univie.ac.at; 2Plant-Insect-Interactions, Department of Life Science Systems, School of Life Sciences, Technical University Munich, 85354 Freising, Germany

**Keywords:** mouthparts, preoral cavity, liquid uptake, filtration, Vespidae, Hymenoptera, µCT, SEM

## Abstract

**Simple Summary:**

Adult Yellowjackets only take up sugar-rich liquid for their daily diet. As these fluids are usually collected from sources with high amounts of various particles, such as fruit flesh or shards of bark, it is important to filter these particles, especially as these animals have a wasp waist with a constriction of the gut, through which the ingested fluid has to pass. We conducted experiments with variously-sized glass particles that were provided in a sugar solution. Female workers of the German Wasp were X-rayed after the food uptake. Possible areas of filtration were investigated with scanning electron microscopy to visualize the surface structures. We could identify two possible areas with filtration function. The first is at the frontal part of the mouthparts, at which the animals could prevent the uptake of particles that were larger than 0.2 mm. A second area inside the head with rows of interlinking hair-like structures enables much finer filtration. Particles that were bigger than 0.15 mm were reliably filtered. The particles that were filtered at this second filtration area were stored in a pouch-like organ inside the head. Smaller particles were also filtered, but some of these were able to pass. These results demonstrate fluid filtration, a poorly-studied topic of insect morphology and behavior.

**Abstract:**

The mouthparts of Vespidae have evolved to forage various solid and liquid foods, such as animal prey, carbohydrate-rich fluids, as well as woody fibres for nest construction. Before nutritional fluids are ingested into the crop, bigger particles need to be filtered out. This study examined the functional morphology of the mouthparts, the preoral cavity, and the proximal alimentary tract inside the head focusing on this filtration process. The feeding organs and preoral cavity were studied using µCT and SEM that were complimented by feeding experiments with glass beads in workers of *Vespula germanica*. To visualize fluid ingestion into the head and alimentary tract, barium sulfate solution was used as contrast agent; a method that is rarely applied in entomology. Experimental results indicate that large glass beads (>212 µm) were filtered by the mouthpart structures before entering the preoral cavity. Smaller glass beads (152–212 µm) were found inside the infrabuccal pocket in front of the mouth. Morphological evidence indicates that cuticle structures of the epipharynx, hypopharynx, and cibarium filter this particle size inside the preoral cavity while glass beads < 152 µm reach the crop. A double fluid filtration system is proposed that is formed by (1) bristles of the mouthparts and (2) microtrichia of the preoral cavity.

## 1. Introduction

Various kinds of liquid food are the primary food source of most adult Hymenoptera. Their mouthparts evolved various specializations for fluid uptake and often form a proboscis for the extraction of floral nectar in flower-visiting taxa [1,2]. Many Hymenoptera evolved a more or less elongated labio-maxillary complex that functions either as a licking organ or a siphoning proboscis [1,3,4]. Biting-chewing mandibles combined with licking mouthparts occur in all eusocial Vespidae that are used for diverse foraging activities (e.g., hunting and scavenging of animals) as well as for liquid uptake [3,5,6,7]. Using the mandibles, female social Vespidae collect animal proteins for larval provisioning and woody fibers for nest construction [5,8,9,10]. With the labio-maxillary complex, they collect water and carbohydrate-rich fluids [5,8,9,10]. Sugary fluids e.g., tree sap, honeydew, nectar, and ripe fruits are important resources for daily activities of the adult wasps and are needed for the development of a colony [6,9]. Similar to water, the carbohydrate-rich fluids are taken up with the apical labial components, ingested, stored in the crop, and shared amongst other adults and larvae by trophallaxis in the nest [5]. In addition, female workers collect animal prey and the parts of carcasses mainly for their larvae [5]. The prey is killed by using the mandibles [5,9]; afterwards the thorax is usually cut open, malaxated, and carried to the nest [6,9]. Inside the nest, the prey is further malaxated to a specific piece size before the meat-bolus is positioned on the mouthparts of the larvae, in appropriate sizes [5,10,11]. During malaxation, the animal fluids are partly ingested by the female workers which is a protein source for adults [5,11]. The other protein source for the adults are droplets of salivary fluid that are received from the larvae via trophallaxis; in this case, the fluid is exchanged between wasp and kin [3,5,10,11]. All kinds of nutritional fluids have to pass the narrow aculeate waist between the first and second segment of the abdomen before digestion in the midgut. Also, as semi-liquid fluids from various resources are taken up, the impurities and particles occur that might block this passage. Therefore, it was assumed that particles are filtered out in some way by the mouthparts [10,12,13].

The morphology of wasp mouthparts is well studied in various species [8,14,15]. Adult Vespinae and Polistinae have stout and heavily sclerotized mandibles with a four-toothed cutting edge [3,8,14,16]. The labio-maxillary complex is a functional unit of the maxilla and labium that is hinged together with the labio-maxillary jugum, a sclerite of the submentum [4,8,17,18]. The maxillae consist of a slender cardo, the flat, triangular stipes that bears the six-segmented maxillary palpus as well as the lacinia and, on the lateral side, the galea [8]. Both the lacinia and galea are less sclerotized than the cardo and stipes but bear numerous long setae and bristle-shaped sensilla [8,14]. The labium can be divided into the submentum, mentum, and prementum. Its distal component, the ligula, is composed of the unpaired lobular-shaped glossa and paired lateral paraglossae [8]. When feeding, the glossa and paraglossa are the first organs, which contact fluids. Both components are equipped with specialized spatula-shaped microtrichia on the anterior side, which are important for liquid transport during licking movements, that are present in many hymenopteran species [19]. The apical acrosomal buttons are smooth areas that bear sensilla basiconica probably with chemosensory function [8,14,15].

With a closer look at the components of the preoral cavity and the fluid passage to the mouth opening, we investigated workers of eusocial wasps that ingested fluids when particles were suspended in the liquid. For the uptake of fluids, adult Vespinae and Polistinae use the protruded labio-maxillary complex in an arrangement that is termed “Wespenrüssel”—wasp proboscis [1,3,8,14,15]. The labial glossa and paraglossa are extended to the front, liquid adheres to it, then it is retracted to transfer the liquid to the proboscis [8,15,20,21]. The fluid-channeling organs of the preoral cavity are frontally formed by the epipharynx and the basal maxillae while the posterior cover is primarily formed by the hypopharynx [8,22,23]. As nutritional fluids are transported further into the head by the cibarium and pharyngeal pump, they pass the functional mouth opening, that is marked by the oral pecten, which opposes the infrabuccal pocket, a unique organ of the Hymenoptera [4,5,8,15,24]. The infrabuccal pocket is formed by an invagination of the hypopharynx; it has globular shape in many Apocrita and can regularly be observed to be filled with solid matters in Vespinae, similar to Formicidae [5,8,15,25,26].

Adult aculeate Hymenoptera, in general, can only ingest fluids or semi-fluid nutrients due to their narrow wasp waist [10,12,13]; the filtration of nutritional fluids must be expected since particles could block this narrow part of the gut. The oral pecten or the functional mouth opening were hypothesized to play a role in the filtration process [5,8,27]. Likewise, the infrabuccal pocket, usually containing detritus and other solid particles [8,12,15], could be functionally important in filtering liquids. The few reports of the filtration process are vague in wasps since it is unknown how particles are expelled, which size of particles can pass the preoral cavity, or are collected in the infrabuccal pocket of wasps.

The aim of this study is to analyses how wasps react to particle contamination of liquid food. In a non-destructive approach, we experimentally tested if workers of *Vespula germanica* are capable of separating various-sized particles from ingested fluid. In addition, the morphology of the preoral region was studied in detail to search for possible filtering structures. This focus on the functional aspects sheds new light on the hymenopteran mouthpart morphology

## 2. Materials and Methods

Female workers of *Vespula germanica* (*n* = 62) were caught at the Naschmarkt, a fruit market in Vienna (Austria) in the summer of 2019 and 2020. They were transferred to the lab and stored in a refrigerator at 5–7 °C (several hours up to 24 h) to immobilize them. After sufficient cooling, they were put into a half of a plastic pipette-head, which had been closed using an adhesive tape to cover the insect except the head. The wasp was then put in close proximity to a small plastic tube that served as a presentation table for the liquid drops. This gave access to the presented liquid food, similarly as it is done with honeybees for proboscis reflex conditioning [28].

The feeding experiments were filmed with a Nikon D5300 and a D500 (Nikon Corporation, Minato, Tokyo, Japan) camera and a Sigma 105 mm F/2.8 EX DG Macro OS HSM (Sigma Corporation, Kawasaki, Japan). In some cases an AF-S NIKKOR 50 mm 1:1.4 G (Nikon Corporation) was used with a fixed open aperture reversed on the Sigma lens. The Nikon cameras were used in video mode. Single frames were processed using Lightroom 6.1 and Photoshop CS (Adobe Inc., San Jose, CA, USA). Drops of sugar solution (in a concentration of approximately 30%) were presented on the presentation table. The wasp could take up the fluid ad libido from several drops. Different solutions and mixtures were presented to the female workers of *Vespula germanica*. Mixtures of 30% sugar water with glass beads were presented. A total of 1 mL of sugar water that was mixed with 0.25 mL glass beads in various sizes was used (Table 1).

In addition, mixtures of 30% sugar water plus glass beads and 2% barium sulfate solution were presented to the wasps. Barium sulfate solution is regularly used as a standard contrast agent in humans for X-ray and CT scans [29], but very rarely in insects [30]. After feeding various glass bead sizes (Table 1), the wasps were either put immediately into a fixative, a mixture of formaldehyde-acetic acid-alcohol (FAA 10:3:1) or were stored 30–60 min in the refrigerator before fixation. After fixation for several days the specimens were transferred to 70% ethanol in which they were stored.

X-ray images were used to evaluate the outcome of the feeding experiments. These X-ray images were photographed from various angles by rotation of the animals at the longitudinal axis, similar to a µCT. The wasps were mounted upright in a plastic tube. This was done “wet” as the animals were taken out of 70% ethanol, and the liquid adhering on the surface was not removed, to ensured that the animals did not dry out during the imaging. A X Radia Micro XCT (University of Vienna, Department of Evolutionary Biology) and a Skyscan 1272 (Bruker, Kontich Belgien, University of Vienna, Department of Evolutionary Biology) were used for the X-ray investigations. The X-ray images were evaluated manually, and the position of glass beads was noted in three categories:None: No glass beads were visible inside the head, thorax, or abdomen. If glass beads were located on the mouthparts, it also was counted as not inside the head.Inside the infrabuccal pocket: Glass beads were visible inside the head as globular shadows in the infrabuccal pocket. The mandibular joint was used to determine if a glass bead was inside or outside the head, as it was viewed from a lateral direction, in a close position to the functional mouth opening.Abdomen: Glass beads were visible inside the anterior abdomen, where the crop is located.

If an animal had taken up glass beads into the preoral cavity of the head and into the abdomen, it was categorized as both (2) inside the infrabuccal pocket and (3) inside the abdomen. In one specimen (<152–212) the abdomen couldn’t be evaluated. It was still included in the given data, as the head was evaluable. To support the evaluation of the maximum size of the glass particles which can be filtered with the mouthparts, a Chi-squared contingency table was used with glass beads presence and absence in both, the infrabuccal pocket as columns and the glass bead sizes as rows. The Chi-squared statistics were calculated using a Fisher’s exact and a Cramer’s V using PAST, Version 4.06b [31].

For 3D analysis of the cuticular components, µCT images of *Vespula germanica* were used. The worker was fixed in FAA and stored in 70% ethanol until scanning. To enhance the contrast, the individuals (*n* = 2) were dehydrated with an ascending ethanol series and stained with 1% iodine solution for at least 1 day [32]. The X Radia Micro XCT was primarily used for the µCT scan. The scan was segmented in AMIRA (Version 6.1, Thermo Fisher Scientific, Waltham, Massachusetts), with the aid of Biomedisa [33]. The surface was calculated using AMIRA and, after a reduction and smoothing, was transferred to Dragonfly (Version 2021, Object Research Systems, Montréal, Canada) for the final rendering. The heads and mouthparts of two female workers of *Vespula germanica* were studied to search for putative filtering structures. The two different dissections were used to view the internal organs of the preoral cavity.
The head was removed from the body and the antennae were cut off. Then, a sagittal cut was performed using a stiff razorblade.The head was removed and a parasagittal cut close to the mandibular joint was performed. The musculature and cuticle structures such as the tentorium were removed, until it was possible to extract the pharynx with the infrabuccal pocket plus hypopharynx and epipharynx. The dorsal and ventral parts were separated carefully with the use of forceps, dissection needles, and microscissors.

The SEM work was performed at the Core Facility Cell Imaging and Ultrastructure Research (CIUS), University of Vienna. After dissection, the specimens were dehydrated with increasing ethanol concentrations (30 min 80%, 45 min 90%, 1 h 96%, 3 times 1 h 100%) on a shaker. The parts were subsequently critical point dried in a Leica CPD 300 (Leica Biosystems, Wetzlar, Germany) with the program: 50%, auto, CO_2_ in (slow, 2 min delay), exchange (speed 2, cycles 30), and gas out (head med., speed slow 100%) was used. The specimens were mounted on aluminum stubs using sticky carbon tabs and silver paste. The stubs were sputter-coated with gold with a JEOL JFC-2300HR (Jeol ltd. Tokyo, Japan) for 120 s. The SEM analysis was performed with an FEI (Philips) XL 30 ESEM (Thermo Fisher Scientific Inc., Waltham, MA, USA) using different angles, spot sizes, and voltages. The digital measurements were done on the resulting images with Fiji (v1.53c) [34].

The terminology has changed in the last century substantially and many older publications on wasp morphology with varying terms are very relevant until today. Therefore, we decided to include a set of terms and definitions that was based on the web portal HAO in the Appendix A to clarify any uncertainty.

## 3. Results

The preoral cavity is a cavity formed by the epipharyngeal wall and hypopharynx, and encompasses a substantial space between the mouthparts and the alimentary canal (Figure 1A,B). The ventral part of the preoral cavity, the epipharyngeal wall, can be further separated in the distal epipharynx or epipharyngeal lobe proximal part of the epipharyngeal wall, which has no specific name, and, therefore, is further referred to as the “posterior” epipharyngeal wall. This part bears close to the functional mouth opening a row of setae, termed oral pecten (Figure 1B). The posterior cover of the preoral cavity is provided by the hypopharynx (Figure 1B). This can also be subdivided from ventral to dorsal into the distal hypopharynx, which opposes the epipharynx, the infrabuccal pocket, the hypopharyngeal lobe, and the most proximal, the sitophore (Figure 1B). The space between the functional and anatomical mouth opening is termed cibarium.

### 3.1. Fluid Filtration

The feeding experiments with sugar water that was mixed with variously sized glass beads demonstrated the filtrating function of the mouthparts (Figure 2). To take up fluid, the worker extended the labio-maxillary complex. This was achieved by licking movements whereby the glossa was wetted and subsequently retracted into the preoral cavity or, alternatively, the tip of the wasp proboscis was dipped into the liquid and the fluid was sucked in. During this process, glass beads that were bigger than 212 µm were prevented from entering the wasp proboscis (Figure 2B). Smaller particles were taken up into the wasp proboscis although some were collected between the mouthparts and disgorged after the wasp has stopped fluid uptake (Figure 2A).

After treatment with barium sulfate solution (with or without small glass beads), the X-ray-studied *Vespula germanica* showed clear visible shadows in the alimentary tract of their body. The position of both barium sulfate and the glass beads could be well determined (Figure 3). The barium sulfate is recognizable by the cloudy darker shadows inside the organs and additionally stained the interior cuticle, while the glass beads were recognizable by a grainy shadow. When the barium sulfate solution that was mixed with the glass particles was fed, a visual identification of the glass beads and liquids was possible in the X-ray images. Barium sulfate was taken up and passed into the midgut, while the glass beads were visible in the infrabuccal pocket and the crop. The glass beads of 152–212 µm in size were largely collected in the infrabuccal pocket (Figure 3). No difference in the separation of glass beads and barium sulfate could be detected due to the different waiting times, but when the animals had a longer take up period (15–30 min), the barium sulfate was more prominent in the posterior digestive tract than in the crop.

When testing the different size categories of glass beads, a highly significant difference between the presence and absence of glass beads in both the infrabuccal pocket (χ²(5) = 51.516, *p* < 0.001, φc ≈ 0.912, *n* = 62) and the crop (χ²(5) = 24.428, *p* < 0.001, φc ≈ 0.633, *n* = 62) was found (Table 2). In all wasps that were fed with particles <152 µm, the particles were found in the infrabuccal pocket, and in ~3/5 (58.8%) of the individuals these glass beads were also detected in the crop (Table 2).

Whenever glass beads > 152 µm were fed, none were found in the crop (Table 2). In experiments with the glass bead size range of 152–212 µm, in six out of ten trials, the glass particles were found in the infrabuccal pocket. In all the experiments with even bigger glass beads, no particles were observed in the infrabuccal pocket (Table 2). If the glass beads were mixed with the smallest category, glass beads could be found in the crops as at least the fraction of small glass beads (<152 µm) could be swallowed (Table 2).

### 3.2. Filtering Structures

The nutritional liquid adheres to the spatula-shaped microtrichia of the glossa and paraglossae during licking movements (Figure 2B). The retraction the glossa and paraglossae transports fluid into the “wasp proboscis” that is formed by the mouthparts. The fluid is further transported into the preoral cavity and passes the functional mouth opening, before it can be further transported through the anatomical mouth opening into the alimentary canal (Figure 4A). This first part of the preoral cavity is frontally formed by the epipharynx and posteriorly by the distal hypopharynx (Figure 5A,B). Liquids have to pass the oral pecten, which is positioned at the functional mouth opening and form a sieve-like structure (Figure 5C–E). Further proximal are at both sides of the cibarium hair-like microtrichia which are interlinking (Figure 5E,F). After passing these structures, the liquid can be transported to the proximal end of the preoral cavity that is formed by the anatomical mouth opening (Figure 4A). The liquid then continues to flow through the pharynx, into the oesophagus to the crop where the small glass beads could be detected in the X-ray images (Figure 3A). 

For a tight seal of the distal preoral cavity, we could identify several structures that were discussed by Duncan (1939), the epipharyngeal bar (Figure 5D) that is connected to the lacinial bar. In addition, the posterior connection between the maxilla and the hypopharynx is formed by the galeal pecten (Figure 4B) and the hypopharyngeal pecten (Figure 4C).

#### 3.2.1. Epipharyngeal Wall

The epipharyngeal wall proximal of the epipharynx is covered with rows of dorso-ventrally-oriented setae and microtrichia (Figure 5C–F). At the transition of the epipharynx to the epipharyngeal wall are socketed setae, that are termed oral pecten, with a length of 69 µm (*n* = 1, mean of 10 measurements) and a thickness of 2.7 µm (*n* = 1, mean of 10 measurements) (Figure 3C–E). These oral pecten are easily distinguishable from microtrichia, as they have a disc-shaped base with full-sized setae attached (Figure 5E). Dorsal of the oral pecten are rows of long cuticle plates which are equipped with numerous comb-like microtrichia that protrude on the ventral-anterior side of these plates (Figure 5C,E). These cuticle structures are pointed and measure 33 µm (*n* = 1, mean of 5 measurements) in length and have a thickness of 1.3 µm (*n* = 1, mean of 5 measurements). They are bent towards the functional mouth opening, similar to the oral pecten (Figure 5C). In the first quarter of the “posterior” epipharyngeal wall are two lateral patches of bristle-shaped sensilla that are implemented in the rows of microtrichia (Figure 5D,E). There are two lateral grooves that form a funnel that are found at the transition to the most proximal longitudinal part of the epipharyngeal wall (Figure 5C). Here, the cuticle structure changes from rows of microtrichia to cuticle plates with much shorter, anterior-facing, spine-like microtrichia (Figure 5F).

#### 3.2.2. Infrabuccal Pocket

The infrabuccal pocket (Figure 5A,B) is a soft hollow organ measuring 329 µm in diameter (*n* = 1, Ø of 9 measurements), that is surrounded by a thin epithelium. A transversal slit forms the entrance into the infrabuccal pocket, which is as wide as the hypopharynx. A transversal lip-like formation, which we termed the hypopharyngeal lip, occurs at the transition between the hypopharynx and the infrabuccal pocket (Appendix A) and (Figure 5B). The cuticle area that is posterior to this lip is covered by strong hair-like microtrichia that are similar to those of the epipharyngeal wall and hypopharyngeal lobe (Figure 6A,B), while the ventral microtrichia of this duct are shorter than those on the dorsal side (Figure 5B).

The hypopharyngeal lobe is covered with microtrichia that are similar to the epipharyngeal wall (Figure 6B). Their alignment follows the form of the rounded surface and point towards the infrabuccal pocket. Close to the sitophore, the microtrichia are gradually reduced in size (Figure 6). There are two lateral patches of secretory pores of the hypopharyngeal gland that are present at the dorso-posterior end of the hypopharyngeal lobe (Figure 6B).

#### 3.2.3. Cibarium and Sitophore

The cibarium, ranging from the functional mouth to the anatomical mouth opening showed two distinct areas. The ventral area with a dense microtrichia coverage and the posterior part with smoother cuticular plates. The latter smooth area is best visible at the proximal part of the hypopharynx, the sitophore, which has a triangular or funnel shape starting from the wide stretched and narrow ventral part of the cibarium (Figure 6C). Two prominent and elevated patches of peg-shaped sensilla are visible on the posterior end of the sitophore, close to the anatomical mouth opening (Figure 6C,D).

## 4. Discussion

The filtration of nutritional fluids in Vespidae has been studied extensively for the first time. The filtration of particles has been assumed repeatedly in Vespidae [8] but in contrast to Formicidae, it has not been tested experimentally [27,35,36]. In this study, we could show that two separate filtration steps are performed during liquid uptake in wasps.

### 4.1. First Filtration Step

Regardless of the presence of particles, liquids are taken up by licking movements of the labio-maxillary complex or by the extended, motionless glossa that is directly dipped into the fluid, similar as in Formicidae [1,8,14,15,37]. When the fluid passes the opening between the galeae and hypopharynx, glass beads that are bigger than 212 µm are prevented from entering the preoral cavity (Figure 4A). In this first step, the galeae pecten and bristles of the anterior ligula plate are assumed to be the important structures for the filtration of the rather big particles in front of the preoral cavity [8].

The necessary suction force for fluid ingestion is provided by the expansion of the cibarium by dilator musculature and the pharynx with the pharyngeal dilator [8,21]. The latter muscles dilate the pharynx, which produces the sucking force that probably transports liquid in a straight path from the mouthparts towards the suction force-producing organs [8,21]. In order to avoid any spillages of fluids during this process, several sealing structures are present. When the labio-maxillary complex is extended and folded to the “wasp proboscis”, the epipharynx connects to the maxilla via the epipharyngeal bar and the lacinial bar. The maxilla connects to the hypopharynx via rows of cuticle structures, the galeal pecten, and the hypopharyngeal pecten but also smooth lateral surfaces are pressed together to probably ensure a tight seal [8]. When this proboscis formation is extended, the hypopharynx is possibly concavely depressed which further enlarges the volume of the preoral cavity.

This first filtration step is especially important when rasping fruit flesh or sucking the haemolymph of captured prey, as bigger particles are easily present during feeding from these food sources that could disrupt the fluid uptake [5,6,7,9,10]. Structures that are similar to the galeal comb to avoid or control the intake of bigger particles are not only found in Vespinae, but also in other Vespidae, e.g., the highly specialized Masarinae and can probably found in many Apocrita [38].

### 4.2. Second Filtration Step

As the liquid is transported through the preoral cavity, it passes the functional mouth opening where a second filtration step takes place. The present experiments showed that glass beads (152–212 µm) were detectable in the infrabuccal pocket. The cuticle structures near the functional mouth opening probably filter out the indigestible particles and transport them into this globular-shaped hollow organ (Figure 4A). This second filtration step of ingested nutritional fluid might be crucial to avoid blockages of the alimentary canal at its narrow section that is close to the wasp waist [5,8,12].

The experimental results indicate that even when the smaller glass particles could not be reliably filtered, a larger amount was still prevented from entering. It seems important to avoid higher quantities of indigestible particles, as these could clump and form blockages in the alimentary canal [12,27].

The results in ants are similar since particle sizes of 200–300 µm have been expelled by the mouthparts of *Camponotus pennsylvanicus* [27]. Particles that are smaller than 150 µm were only found in the infrabuccal pocket and smaller than 100 µm were found in both the infrabuccal pocket and the crop. This is similar to the presently-tested workers of *Vespula germanica,* although they are bigger in body size. We assume that the second filtration mechanism is the same in wasps and ants. The much smaller ants, *Acromyrmex octaspinosus* and *Solenopsis invicta* were found to filter even smaller sizes of 0.88–10 µm, however this could be caused by the small body size of the insects [35,36]. It is also reasonable that the different lifestyles might explain the differences between the various Formicidae [27,35,36].

The present morphological results in *Vespula germanica* led to the following functional interpretation. The epipharyngeal oral pecten can be assumed to be the primary and most important filtration structures, as has been discussed in the past [5,8,12,15]. The microtrichia of the epipharyngeal wall and hypopharyngeal lobe are directed against the current of the inflowing fluid and towards the infrabuccal pocket. It is likely that, in various positions, these cuticle structures interlink with each other, forming a kind of sieve which would assist the oral pecten with filtration. Vespidae are presumably able to move the hypopharyngeal lobe in the dorso-ventral direction, which would allow them to transport the filtered particles from the oral pecten and the surrounding structures into the infrabuccal pocket. During these up and down movements, the particles could be combed ventrally towards the entrance of the infrabuccal pocket, which itself is similarly covered with rows of microtrichia, that are likewise directed into the infrabuccal pocket. The orientation of these hair-like microtrichia would push further into the infrabuccal pocket.

This would not only allow the insect to gather indigestible material that are bigger than 152 µm outside the alimentary canal and avoid blockages, but also keep the filtration structures clean and functional during continued feeding. This movement is probably performed during each pumping cycle of the cibarium and pharynx [4]. It can be assumed that the microtrichia support the filtration of particles similar as discussed for the infrabuccal entrance in Formicidae [26]. The shape of the cibarium also further helps to underline this hypothesis, as the fronto-posterior narrow and laterally wide shape of the ventral part of the cibarium is very useful for such a filtration process as it forms a large surface area.

### 4.3. Function of the Infrabuccal Pocket

In both, the Formicidae and Vespidae, this pouch-like organ is used to gather, compress, and glue together the particles prior to being expelled as pellets that are termed “corpuscules de nettoyage” by Janet (1905) [8,15,25]. The cuticula of the infrabuccal pocket in *Ectomomyrmex javanus* (Formicidae) shows different sized hairs [26]. These hairs are oriented towards the opening of the infrabuccal pocket and the size reduces from the entrance into the pocket [26]. A similar endowment with microtrichia could be observed in the studied wasp *Vespula germanica* (Figure 5B).

It has been previously discussed that the infrabuccal pocket in Formicidae is used for filtration and liquids, which are first transferred into the infrabuccal pocket before they further pass into the cibarium [26,39]. The present results in Vespidae do not support this assumption, despite the fact that wasps show a similar arrangement of the cibarial area. The arrangement of the microtrichia, especially at the epipharyngeal wall, rather indicates that filtration is performed at the oral pecten. A guidance of the filtered particles away from the functional mouth opening and storage of particles inside the infrabuccal pocket has been briefly discussed before in wasps [5], but also in ants [35]. As fluids are transported by the suction force into the pharynx, it seems unlikely that a major portion is guided towards the infrabuccal pocket. When wasps were fed with the liquid contrast agent, only small quantities of it were found in the infrabuccal pocket but large amounts stained the alimentary gut posterior inside the body. If nutritional fluid is filtered inside the infrabuccal pocket, it should be expected that more barium sulfate enters into the infrabuccal pocket. We detected no musculature which could suck liquid into the infrabuccal pocket. Therefore, it is unlikely that it is used actively for filtration during feeding as it has been assumed [26,39]. This lack of musculature raises the question of how the particles are transported out of the infrabuccal pocket after feeding. Many reports of expelled pellets are known, suggesting that they are compressed into a tight pellet, possibly under the use of a glandular substance [12,35]. As no muscles are attached to the infrabuccal pocket, we assume that this is performed by the use of haemolymph pressure. A combination with a movement or position of the labio-maxillary complex also seems likely as this organ is an invagination of the hypopharynx and, therefore, functionally connected to other mouthpart components.

### 4.4. Staining and Testing Methods

Barium sulfate is a standard contrast substance in medicine. We used this method to visualize fluid transport in the gut of insects. It could be demonstrated that nutritional fluid is ingested into posterior parts of the alimentary tract but particles are filtered by the mouthparts and cuticle structures of the preoral cavity. The use of glass beads and barium sulfate in our feeding experiments proved to be an efficient and affordable way to stain the alimentary tract and to track particles inside the alimentary tract. The biggest problem with this method is that the bigger glass beads were sedimenting fast in the experimental solution. To counter this, only single drops were presented to the wasps, which ensured that they pushed their mouthparts into the suspended glass beads. The recognition of the glass beads inside the body was easy and clear. Even the separation of glass and barium sulfate was easy in most cases, as the structure of both was clearly different in the X-ray images. The barium sulfate stained the alimentary canal at least for the study period during experiments. There was no indication that the wasps recognized barium sulfate in the sugar solution, as no avoidance behaviour was observed. Overall, the testing of filtration properties with glass beads as a substitute for solid objects in nutritional fluid together with the use of barium sulfate as contrasting substance seems to be an easy-to-use method to study the uptake and transport of fluids that are contaminated by particles.

## 5. Conclusions

We showed that *Vespula germanica*, a common eusocial wasp, is able to filter solid particles out of nutritional liquids. These insects employ two separate filtration steps. The first step occurs at the entrance of the mouthparts where the “wasp proboscis” prevents particles that are bigger than 212 µm from entering the preoral cavity. The second step occurs at the functional mouth opening, reliably filtering particles that are bigger than 152 µm. Smaller glass beads were also filtered, but not reliably, as these often were found in the crop.

Morphological adaptations for filtration are present, especially at the second filter system consisting of the oral pecten and microtrichia as a filtration system inside the preoral cavity. The microtrichia probably help filtering and transporting particles into the infrabuccal pocket, which serves as a receptaculum to store the filtered particles. Therefore, the infrabuccal pocket fulfils a storage function, as the ingested glass beads were accumulating as they were separated from the liquid food. The mechanism of emptying the infrabuccal pocket is still unclear since no appropriate muscles could be found.

The mouthparts of aculeate Hymenoptera, especially of those taxa which are digging, handling plant parts, or predate, need a protection against particles to avoid blockages of the thin alimentary canal. It can be assumed that all aculeate Hymenoptera face this problem and this might be the reason why all of them feed on fluids and most of them can be observed as nectar feeders.

## Figures and Tables

**Figure 1 insects-13-00185-f001:**
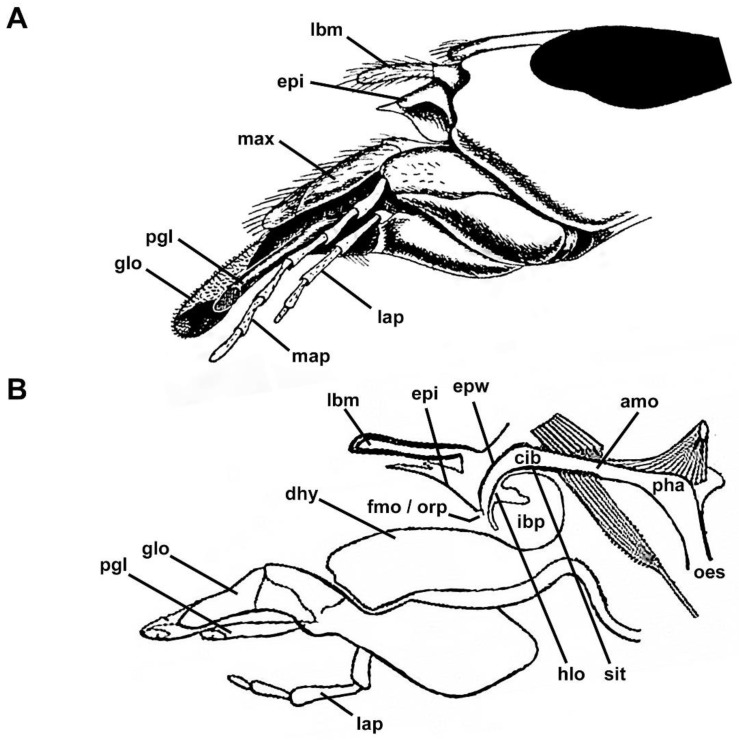
Schematic drawings of the Vespinae mouthparts and preoral cavity. (**A**) Lateral view at the head and mouthparts, without the mandible, edited from Kirmayer (1909). (**B**) Lateral view at the internal anatomy of the preoral cavity, edited from Duncan (1939). [Abbreviations: amo: anatomical mouth opening, cib: cibarium, dhy: distal hypopharynx, epi: epipharynx (distal part of the epipharyngeal wall), epw: epipharyngeal wall, fmo: functional mouth opening, glo: glossa, hlo: hypopharyngeal lobe, ibp: infrabuccal pocket, lap: labial palpus, lbm: labrum, map: maxillar palpus, max: maxilla, oes: oesophagus, orp: oral pecten, pgl: paraglossa, pha: pharynx, sit: sitophore].

**Figure 2 insects-13-00185-f002:**
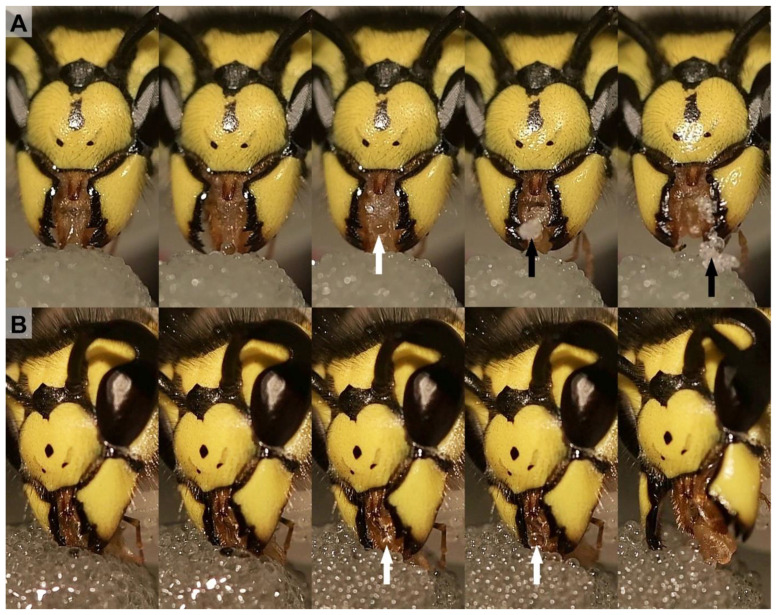
Feeding experiment using glass beads in sugar water with *Vespula germanica*; single frames from the video footage; (**A**) Uptake of the sugar water with the glass beads (<152–600 µm). (**B**) Uptake of the sugar water only with bigger particles (212–300 µm). The bigger glass beads were prevented from entering the wasp proboscis (white arrow). The small glass beads entered between the mouthparts (black arrow); some were filtered out and ejected after the liquid ingestion.

**Figure 3 insects-13-00185-f003:**
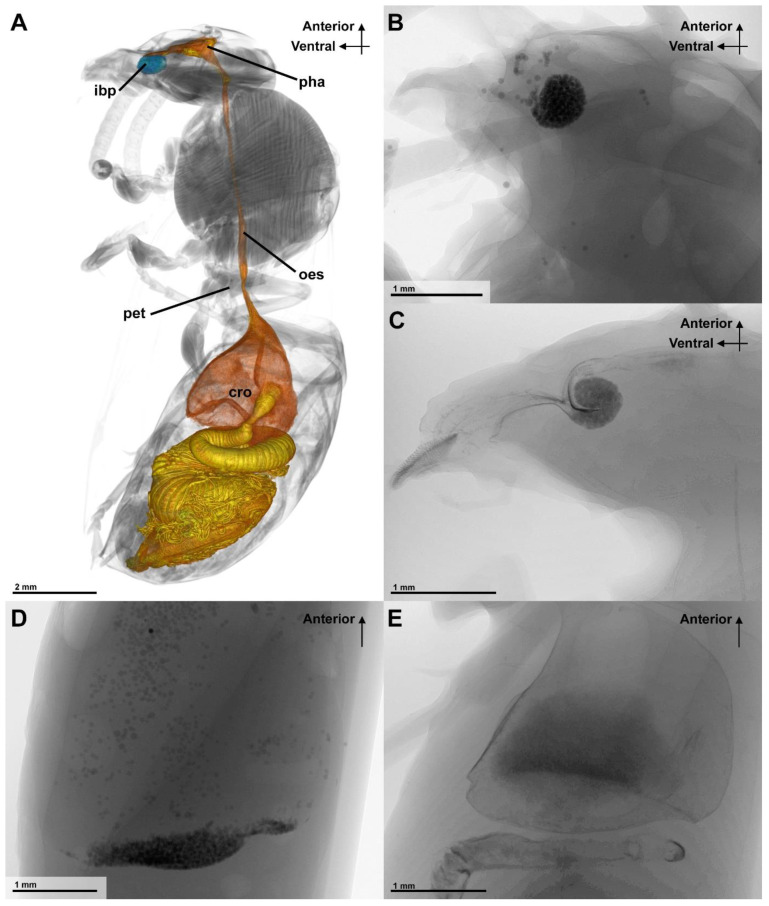
The digestive tract of *Vespula germanica* and the positions of the glass beads after feeding experiments (lateral view). (**A**) The digestive tract (orange, yellow) and the infrabuccal pocket (blue) in micro-CT, volume rendering. (**B**) The glass beads in the infrabuccal pocket after receiving sugar water with glass beads < 152 µm. (**C**) The glass beads in the infrabuccal pocket and cuticle of the preoral cavity that were additionally stained by barium sulfate. (**D**) The glass beads (<152 µm) inside the crop. (**E**) Barium sulfate solution-staining of the crop with a cloudy shadow [Abbreviations: cro: crop, ibp: infrabuccal pocket, oes: oesophagus, pet: petiolus, pha: pharynx].

**Figure 4 insects-13-00185-f004:**
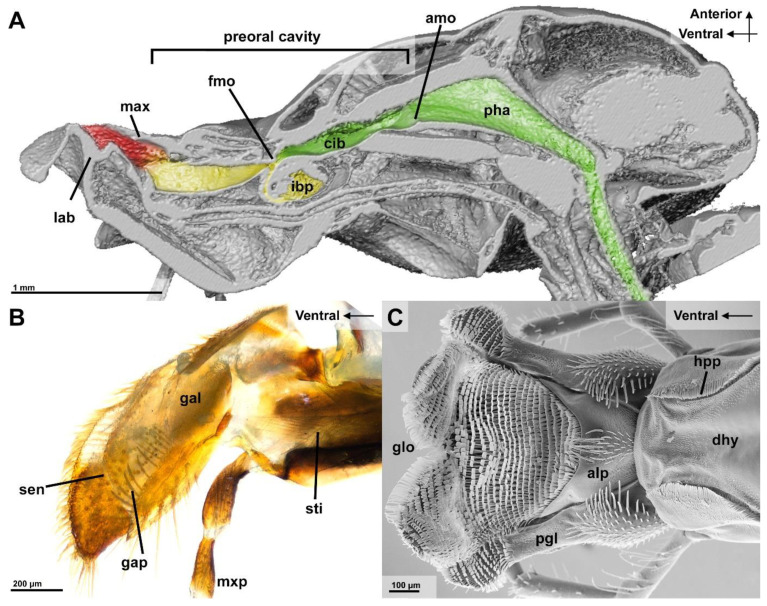
Stages of filtration by the mouthparts and preoral cavity of *Vespula germanica*. (**A**) Surface render of µCT, cut sagittally with the path of liquid colored; (red): unfiltered liquid; (yellow): sizes > 212 µm filtered; (green): sizes > 152 µm filtered. (**B**) Surface detail of the right distal maxillar parts from a medial view (light microscopy). (**C**) The surface of the labium in a dorsal view (SEM). [Abbreviations: alp: anterior ligula plate, amo: anatomical mouth opening, cib: cibarium, dhy: dorsal hypopharynx, fmo: functional mouth opening, gal: galea, gap: galea pecten, glo: glossa, hpp: hypopharyngeal pecten, ibp: infrabuccal pocket, lab: labium, max: maxilla, mxp: maxillary palpus, pgl: paraglossa, pha: pharynx, sen: sensillae, sti: stipes].

**Figure 5 insects-13-00185-f005:**
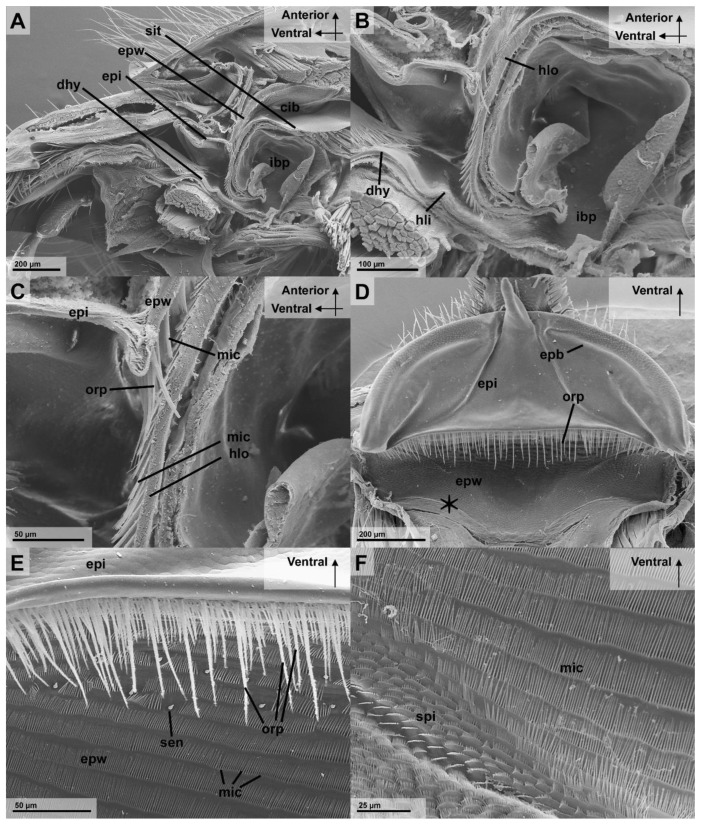
The preoral filtering structures inside the head of *Vespula germanica*, focused on the parts of the ephipharyngeal wall (including the epipharynx) (SEM). (**A**) The anterior parts of the head in a sagittal cut. The preoral cavity and the functional mouth opening (at the oral pecten) is visible. (**B**) The detail of the infrabuccal pocket in a sagittal cut. (**C**) The detail of the arrangement of microtrichia and the oral pecten at the functional mouth opening, lateral view. (**D**) The overview of the epipharynx and epipharyngeal wall, viewed from the posterior-ventral side. (**E**) The detail of the oral pecten from the posterior-ventral view. (**F**) The detail of the surface structure of the posterior epipharyngeal wall, with the cuticula structures changing from hair-like microtrichia to the spine-like microtrichia, at the funnel-shaped lateral groove, marked in (**D**) with an asterisk, viewed from posterior-ventral [Abbreviations: cib: cibarium, dhy: dorsal hypopharynx, epb: epipharyngeal bar, epi: epipharynx, epw: epipharyngeal wall, hli: hypopharyngeal lip, hlo: hypopharyngeal lobe, ibp: infrabuccal pocket, mic: microtrichia, orp: oral pecten, sen: sensillae, spi: spine-like microtrichia].

**Figure 6 insects-13-00185-f006:**
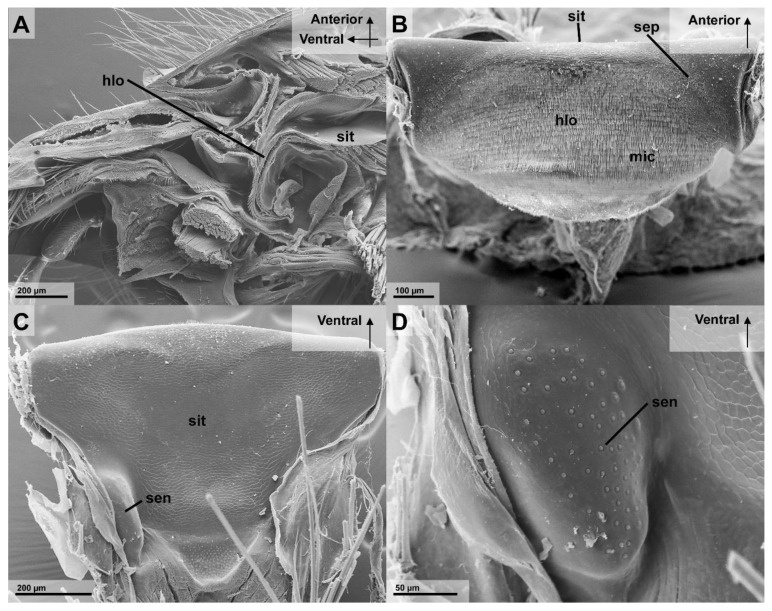
Preoral filtering structures inside the head of *Vespula germanica*, focused on the parts of the hypopharynx (SEM). (**A**) The anterior parts of the head in a sagittal cut. The preoral cavity and the functional mouth opening are visible. (**B**) An overview of the hypopharyngeal lobe, viewed from the frontal side, with microtrichia and a field of secretory pores visible. (**C**) The surface structure of sitophore, viewed from the dorsal side (**D**) The detail of the elevated patch of sensilla at the posterior side of the sitophore. [Abbreviations: hlo: hypopharyngeal lobe, mic: microtrichia, sen: sensilla, sep: secretory pores, sit: sitophore].

**Table 1 insects-13-00185-t001:** Mixtures of 30% sugar water with various glass bead sizes (Sigma, glass beads, acid washed < 152 µm G4649, 152–212 µm G-1145, 212–300 µm G-1277, 450–600 µm G-8772) that were experimentally fed to workers of *Vespula germanica*. In the glass bead category < 152–212 both sizes < 152 µm and 152–212 µm were combined and in < 152–600 all four different glass bead sizes occurred.

Glass Bead Size (µm)	Number of Experiments
<152	*n* = 19
152–212	*n* = 10
212–300	*n* = 13
450–600	*n* = 5
<152–212	*n* = 10
<152–600	*n* = 5

**Table 2 insects-13-00185-t002:** Feeding experiment using *Vespula germanica* workers. Different sized glass beads were fed with sugar water (30%) resulting in differences in the presence (Yes) and absence (No) of glass beads in the infrabuccal pocket and/or crop.

Glass Bead Sizes	Number of Workers	Glass Beads in Infrabuccal Pocket ^1^		Glass Beads in Crop ^2^	
(µm)	(*n*)	Yes	No	Yes	No
<152	19	100%	0%	63%	37%
152–212	10	60%	40%	0%	100%
212–300	13	0%	100%	0%	100%
450–600	5	0%	100%	0%	100%
<152–212	10 ^3^	100%	0%	40%	50%
<152–600	5	100%	0%	60%	40%

^1^ Results of χ² contingency table for glass beads in infrabuccal pocket: χ²(5) = 51.516, *p* < 0.001, φc ≈ 0.912. ^2^ Results of χ² contingency table for glass beads in crop: χ²(5) = 24.428, *p* < 0.001, φc ≈ 0.633. ^3^ The abdomen of one individual was not evaluable.

## Data Availability

Data are available on request from the corresponding author. Animals from the experiments will be stored in the insect collection of the faculty of life sciences (Department for Evolutionary Biology–University Vienna).

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
