# Peer review of "Filtration of Nutritional Fluids in the German Wasp Vespula germanica (Vespidae, Hymenoptera)"

_insects, 2022, doi:10.3390/insects13020185_

Round 1
Reviewer 1 Report
Dear Editor and Authors,
I had read with a real interest the paper concerning the filtration of nutritional fluids in wasp and I was impressed by the used methodology. Indeed, it is an important step in the knowledge of the functional morphology of the MLC and of the anterior part the digestive tract.
In my opinion, the introductive part can be improved framing this work in a broader context – studies on the same line in Hymenoptera. This can be done by citing several extra papers and highlighting the novelty of this present work has (e.g. this paper, unlike the others focused on morphology, explains some morphological structures in terms of their functionality).
Beutel RG, Vilhelmsen L. 2007. Head anatomy of Xyelidae (Hexapoda: Hymenoptera) and phylogenetic implications. Organisms Diversity & Evolution 7: 207–230.
Bugnion E. 1924. Le sac infrabuccal et le pharynx des fourmis. Comptes Rendus des Séances de la Société de Biologie 91: 998–1000.
Bugnion E. 1925. Notes relatives a la terminologie des organes buccaux des insectes. Bulletin de la Société Zoologique de France 50: 352–358.
Bugnion E. 1930. Les pieces buccales, le sac infrabuccal et le pharynx des fourmis. Bulletin de la Société Entomologique d’Égypte 14: 85–210.
Iuga VG. 1968. Le complexe labro-maxillaire des Apoides (Hymenoptera). Travaux du Muséum National d’Histoire Naturelle ‘Grigore Antipa’ 8: 909–926.
Jervis M. 1998. Functional and evolutionary aspects of mouthpart structure in parasitoid wasps. Biological Journal of the Linnean Society 63: 461–493.
Jervis M, Vilhelmsen L. 2000. Mouthpart evolution in adults of the basal, ‘symphytan’, hymenopteran lineages. Biological Journal of the Linnean Society 70: 121–146.
Matsuda R. 1957. Morphology of the head of a sawfly, Macrophya pluricincta Norton (Hymenoptera, Tenthredinidae). Journal of Kansas Entomological Society 30: 99–108.
Matsuda R. 1965. Morphology and evolution of insect head. Memoirs of the American Entomological Institute 334: 1–334.
Popovici O, Miko I, Seltmann K, Deans A. 2014. The maxillo-labial complex of Sparasion (Hymenoptera, Platygastroidea). Journal of Hymenoptera Research 37: 77–111.
Popovici O, Vilhelmsen L, Masner L, Miko I, Johnson N. 2017. Maxillolabial complex in scelionids (Hymenoptera: Platygastroidea): morphology and phylogenetic implications. Insect systematics and evolution 48(4): 315– 439.
Prentice MA. 1998. The comparative morphology and phylogeny of apoid wasps (Hymenoptera: Apoidea). PhD Thesis, University of California, Berkeley, CA.
Snodgrass RE. 1932. Evolution of the insect head and the organs of feeding. Report of Smithsonian Institution for 1931: 443– 489.
Author Response
Dear Reviewer,
Thank you very much for your input. We are happy that the Manuscript could interest you and we appreciate your comments.
"In my opinion, the introductive part can be improved framing this work in a broader context – studies on the same line in Hymenoptera. This can be done by citing several extra papers and highlighting the novelty of this present work has (e.g. this paper, unlike the others focused on morphology, explains some morphological structures in terms of their functionality)."
I can definitely see your point there and tried to improve the citation situation by adding papers mentioned in your list. Sadly in the short time it wasn’t possible to apprehend all of them, especially some of the old papers, and the thesis from Prentice, M.A. 1998.- I could find and include at least one paper per author, sometimes more and added them.
I also found in one of the papers (Popovici et al. 2014) the HAO mentioned, which immediately struck my interest. Another Reviewer advised a clarification of the abbreviation, and used morphological terms, and that was perfect.
I hope the modifications are what you hoped for and of course we would be happy if this paper is also of value for your future research.
Yours sincerely
Kenneth Kuba, Harald W. Krenn
Reviewer 2 Report
Dear Authors,
The manuscript you submitted is an essential contribution to the functional morphology of feeding in Hymenoptera and I strongly support the publication with some corrections to be made. They are mostly related to morphological descriptions and illustrations.
Please read my detailed comments in the attachment.

Author Response
Dear Reviewer,
Thank you very much for your comments, and your recommendation. It is highly appreciated as the comments were of great value to us. We tried to include as many of your suggestions as possible and with your aid, it was possible to improve several aspects of the paper. We added a supplement with a table of the used morphological terms with closer definitions and a list of abbreviations. This should together with the in-text corrections and added pictures help with the confusing situation. We hope the corrections applied are what you hoped for and hopefully the present manuscript can also be interesting for your future work.
The corrections are further explained in the attachment
Yours sincerely,
Kenneth Kuba, Harald W. Krenn

Reviewer 3 Report
This paper examined the functional morphology of the mouthparts of one wasp species Vespula germanica. The filtrating ability of mouthparts was investigated by feeding experiments with various glass beads, and the structure of preoral cavity was observed by using μCT and SEM. In particular, μCT and barium sulfate solution are very novel method in this study.
This study involves complex structure of insect mouthparts, which is difficult to describe, but the author adopted a variety of novel methods to conduct experiments, and obtained good experimental results, which verified the scientific questions proposed.
There are still some minor problems in the paper, which need further modification:
Line 128: description of glass bead sizes in Table 1are different from those in Table 2, such as “<152+152-212” in Table 1 and “<152-212” in Table 2, “<152+152-212+212-300+450-600” in Table 1 and “<152-600” in Table 2. Which is better to use in whole manuscript?
Line 249: In Table 2, in penultimate line, glass beads in crop, yes is 4, no is 5, why isn’t their sum 10?
Line 271-279 (Page 9): Some abbreviations are not consistent in Figure3 and its captions. Such as EPW and EpW are not found in the captions, Mt isn’t found in the Figure. HPL and HpL are same?
Line 322-327: Where is the Figure 4 in the text?

Author Response
Dear reviewer,
Thank you for input. It has been highly valuable and we truly appreciate it. The noted points have been worked on and you will find that some have been completely changed. Hopefully the corrections applied are what you hoped for and ideally the present manuscript can also be interesting for your future work.
The corrections are further explained in the attachment
Yours sincerely,
Kenneth Kuba, Harald W. Krenn
There are still some minor problems in the paper, which need further modification:
Line 128: description of glass bead sizes in Table 1are different from those in Table 2, such as “<152+152-212” in Table 1 and “<152-212” in Table 2, “<152+152-212+212-300+450-600” in Table 1 and “<152-600” in Table 2. Which is better to use in whole manuscript?
I agree that can be confusing. I decided to only use the shorter version, and explain the mixures a little bit more in Table 1
Line 249: In Table 2, in penultimate line, glass beads in crop, yes is 4, no is 5, why isn’t their sum 10?
There has been a problem with one specimens and therefor the abdomen couldn’t be evaluated. This should have been noted, and will be.
Line 271-279 (Page 9): Some abbreviations are not consistent in Figure3 and its captions. Such as EPW and EpW are not found in the captions, Mt isn’t found in the Figure. HPL and HpL are same?
I will completely redo the abbreviation. A other reviewer noted that it is confusing and will change it to a stil similar to Vilhelmsen 1996, with a few additions. I will also include a table with definitions for the used Terms.
Line 322-327: Where is the Figure 4 in the text?
Oh, yes that’s missing. I will definitely refer to it. Thank you!

Reviewer 4 Report
Minor corrections:
Lines 250, 252: please add a space in “,p”.
Line 361: please change “e,g,” to “e.g.”.
Line 364: please change “though” to “through”.
Line 503: please add a coma after “Krenn”.
Line 528: please change “O,T.” to “O.T.”
Line 562: please change “Cham,Switzerland” to “Cham, Switzerland”.
Line 533: “W,H.” should be also changed to “W.H.”? Please check the references list consistently.
Comment on the study:
This study is a beautiful and informative addition to the investigations of the functional morphology of insect mouthparts. It provides new insights into the mechanics of fluid filtration in insects. Investigations of this sort have never been done before, and the experimental part of this study shows originality and thoroughness. Therefore, I totally recommend acceptance and publication of this paper.
It is also nice that the results are being compared to the filtration apparatus in other hymenopterans, which additionally opens a discussion on the filtration mechanisms in insects with different body size.
Author Response
Dear Reviewer,
Thank you very much for your comments. I admire your eye for these details, as I often miss them. I have edited all of your notes, and tried my best to find more of these mistakes. Both of the authors had another detailed look at spelling mistakes, to avoid as much as possible.
Hopefully the present manuscript can also be interesting for your future work.
Yours sincerely,
Kenneth Kuba, Harald W. Krenn